# Enhanced Accuracy of CMOS Smart Temperature Sensors by Nonlinear Curvature Correction

**DOI:** 10.3390/s18124087

**Published:** 2018-11-22

**Authors:** Gareth D. Lewis, Patrick Merken, Marijke Vandewal

**Affiliations:** Signal and Image Centre, Royal Military Academy, Avenue de la Renaissance 30, B-1000 Brussels, Belgium; patrick.merken@rma.ac.be (P.M.); marijke.vandewal@rma.ac.be (M.V.)

**Keywords:** calibration, smart sensors, characterization, curvature correction, temperature sensor

## Abstract

In this paper, we demonstrate an improvement in the accuracy of a low-cost smart temperature sensor, by measurement of the nonlinear curvature correction at multiple temperature references. The sensors were positioned inside a climate chamber and connected outside to a micro-controller via a network cable. The chamber temperature was increased systematically over a wide range from −20 ${}^{{}^{\circ}}$C to 55 ${}^{{}^{\circ}}$C. A set of calibration curves was produced from the best fitting second-order polynomial curves for the offset in temperature between the sensor and reference. An improvement in accuracy of ±0.15 ${}^{{}^{\circ}}$C is with respect to the mentioned temperature range, compared to the significantly higher value reported of ±0.5 ${}^{{}^{\circ}}$C by the manufacturer for similar conditions. In summary, we demonstrate a significant improvement in the calibration of a low-cost, smart sensor frequently used in research and academic projects over a useful range of temperatures.

## 1. Introduction

The measurement of temperature transcends all aspects of our society, from the fundamentals of heat transfer and thermodynamics [1,2], to applications requiring process control, systems protection and calibration [3]. Increasingly, smart integrated circuit (IC) temperature sensors substitute conventional ones such as thermocouples and resistance devices, for those applications requiring measurements in the range from −55 ${}^{{}^{\circ}}$C to 125 ${}^{{}^{\circ}}$C (military range) [4]. Smart sensors are commonly fabricated in complementary metal oxide semiconductor silicon (CMOS), which satisfies the joint requirement for low-cost and high volumes of devices. Specifically, the rapid growth in consumer electronics is a reason for the proliferation of smart temperature sensors that support the thermal management of all electronic systems [4,5].

A smart sensor converts an analogue signal to a digital readable value via circuit biasing and analogue-to-digital conversion (ADC), which, for a temperature measurement, is frequently a bipolar junction transistor (bjt) [4,6,7]. A number of approaches that exploit the temperature sensing capabilities of these transistors are described by Meijer in Ref. [5]. The principal advantage of bjts is the near-linear change in output voltage across the base-emitter terminal of a transistor ($V_{BE}$) that is proportional to the absolute temperature (PTAT). The most promising combine the PTAT measurement of the $V_{BE}$ between two similar transistors compensating for process errors, with a third transistor used as a reference—a ratiometric measurement [4]. This approach is the basis for many smart temperature sensors.

There are two main limitations of smart temperature sensors when compared to conventional ones: (1) the operational temperature range is limited typically to between −55 ${}^{{}^{\circ}}$C and 125 ${}^{{}^{\circ}}$C and (2) the sensors exhibit a mediocre temperature accuracy without additional calibration (0.5–2 ${}^{{}^{\circ}}$C). The former is a constraint of CMOS based circuitry even though these devices could be adapted for operation at higher temperatures with an increased cost [4]. For example, silicon on insulator (SOI) operates up to temperatures of 250 ${}^{{}^{\circ}}$C increasing up to 600 ${}^{{}^{\circ}}$C when the heated region is isolated from the rest of the CMOS circuitry by a dielectric [8]. In spite of these limitations, there are numerous applications for these [9,10,11].

The limitation in temperature accuracy is principally a trade-off between the production costs for calibration against the precision required. In principal, one reference temperature is sufficient during calibration in the foundry, which is typically chosen to be at the midpoint of the operating temperature range. The second point necessary to fix a linear fit of the output voltage with temperature is given by the theoretical bandgap voltage of Silicon at zero Kelvin $\left(V_{g0}=0\right)$. However, not only is there a discrepancy between the theoretical value of $V_{g0}$ predicted from physics and that empirically derived from experiments [4], but the variation of voltage with temperature is typically not perfectly linear. Pertijs et al. [12] demonstrated that an accuracy of ±0.1 ${}^{{}^{\circ}}$C could be achieved by reducing the second-order effects, with circuitry in addition to a one point correction after packaging. A similar calibration accuracy for low-power sensors was obtained by Souri et al. [13]. Batch calibration is also a means to improve the sensor accuracy since devices of the same wafer are likely to exhibit a similar response [14].

Alternatively, one could calibrate at additional reference temperatures in a small climate chamber only for those sensors required for the experiment. Therefore, the user decides whether the increase in accuracy is worth the increased cost. The objective of this paper is to improve the accuracy of a low-cost digital smart temperature sensor (ds18b20) by correcting for the nonlinear response at multiple temperature references. Three key aspects of the sensor performance will be studied in the subsequent sections: (i) accuracy relative to the original claims by the manufacturer, (ii) repeatability of the measurement at a constant reference temperature and (iii) improvement of calibration accuracy by correction of the nonlinear temperature response of the sensor.

In Section 2, we describe materials and methods, including the theoretical background, necessary to explain the need for a nonlinear correction, outline the measurement procedures and explain the experimental apparatus. In Section 3, we present results demonstrating the improvement in accuracy by curvature correction. The paper concludes with discussions on the nonlinear calibration curve (Section 4) and concludes in Section 5.

## 2. Materials and Methods

### 2.1. Curvature Correction

As mentioned previously in the introduction, the base-emitter voltage $\left(V_{BE}\right)$ of a bi-polar transistor cannot be assumed perfectly linear with temperature change. The voltage error at temperatures on either side of the foundry calibration reference $\left(V_{r}\right)$ result in a predictable nonlinear curve, which once determined can improve the calibration accuracy of the sensor. Equation (Equation 1) describes the temperature dependency of $V_{BE}$,(1)$$V_{BE}\left(T\right)=\underset{\mathrm{constant}}{\underset{︸}{V_{BE0}}}-\underset{\mathrm{linear}}{\underset{︸}{\lambda T}}+\underset{\text{higher-order}}{\underset{︸}{c(T)}},$$
where $V_{BE0}$ is the linear extrapolation of the voltage at the reference temperature to absolute zero (T=0 K),(2)$$V_{BE0}=V_{g0}-c\left(0\right),$$
where, $V_{g0}$ is referred to as the bandgap voltage at 0 K [5], λ represents the bandgap voltage and the slope of the tangent, respectively, and(3)$$\lambda =\frac{V_{BE0}-V_{BE0}\left(T_{r}\right)}{T_{r}},$$
as explained in ref. [4] (pp. 51–105) for a reference temperature $\left(T_{r}\right)$ . The impact of curvature on the temperature error is reported in Figure 1a highlighting that the actual voltage measured is always less than the linear approximation except at $\left(T_{r}\right)$, where they are both the same. If the transistor collector current is expressed as proportional to temperature to the *m*th power, we can express the curvature term (c(T)) as Equation (Equation 4),(4)$$c\left(T\right)=\frac{\kappa }{q}\left(\eta -m\right)\left(T-T_{r}-T\ln\left(\frac{T}{T_{r}}\right)\right),$$
where the κ represents the Boltzmann constant, *q* the electron charge, η is a temperature exponent in the analytical expression for saturation current [5], and $T_{r}$ the reference temperature in Kelvin. If derived from a PTAT voltage with a temperature-independent resistor, Equation (Equation 4) is simplified since m=1.

Under the condition of small change between the sensor and reference temperatures, $(T-T_{r})<<T$, we can express Equation (Equation 4) as a Taylor expansion keeping terms only to second order (Equation (Equation 5)),(5)$$c\left(T\right)=\frac{1}{2}\left(\eta -1\right)\frac{\kappa T_{r}}{q}\left({\frac{T-T_{r}}{T_{r}}}^{2}\right),$$
where κ , *q*, and $T_{r}$ are defined previously.

The difference between the voltage at the sensor and reference temperatures is the curvature correction necessary to calibrate the sensors (solid line in Figure 1a). In practice, it is more useful to express the curvature correction in terms of a temperature offset (ΔT), as graphically shown in Figure 1b and below in Equation (Equation 6),(6)$$\Delta T=\frac{T\ln\left(\frac{T}{T_{r}}\right)-\left(T-T_{r}\right)}{1+\frac{qV_{g0}}{\left(\eta -1\right)\kappa T_{r}}}.$$

Additionally, the temperature correction required for curvature increases with η most noticeably when $(T-T_{r})>\pm$25 ${}^{{}^{\circ}}$C (Figure 1b). By rearranging Equation (Equation 6) in terms of curvature, we can show that ΔT∝−c(T). A slight asymmetry of curves in Figure 1b suggests a limitation of the truncation in the Taylor series to only the second-order in Equation (Equation 5). Consequently, knowing the curvature correction enables an enhanced calibration for sensors beyond that of the one reference temperature used in production.

### 2.2. Measurement Procedure

An outline of the measurement procedure is as follows:**Measure** all the output sensor temperatures for each calibration reference temperature.**Calculate** the temperature offset correction between sensor measurements and reference temperatures.**Best fitting** second-order polynomial are calculated from the measured temperature offsets (step 2).**Evaluate** the residual errors at each reference due to the best fitting curve.

### 2.3. Experimental Arrangement

A schematic representation of the experimental arrangement is shown in Figure 2, and images in Figure 3. The principal parts of the apparatus are: climate chamber [15], ds18b20 temperature sensors [16] and an Arduino Yun micro controller [17].

#### 2.3.1. Climate Chamber

The calibration is in a small climate controlled chamber (model WK180/40 supplied by Wiess Technik, Liedekerke, Belgium) that is programmed for a stepped temperature profile from −20 ${}^{{}^{\circ}}$C to 50 ${}^{{}^{\circ}}$C stabilizing in 5 ${}^{{}^{\circ}}$C intervals. At each interval, there are approximately 20 min of stable measurement time. The climate chamber is calibrated with an accuracy of ±0.1 ${}^{{}^{\circ}}$C at two references (23 ${}^{{}^{\circ}}$C and 80 ${}^{{}^{\circ}}$C), which we take as representative of the whole temperature range investigated. Sensor cables were passed to the micro-controller a few meters away, via a small plug-able inlet in the climate chamber.

#### 2.3.2. Smart Temperature Sensors

Fourteen ds18b20 smart temperature sensors were studied [16], with none from the same batch. An example of which is shown in Figure 3a. The sensor is programmable in temperature resolution from 9bits representing an interval of 0.5 ${}^{{}^{\circ}}$C, increasing to the default resolution of 12bits at an interval of 0.0625 ${}^{{}^{\circ}}$C. The default setting was used throughout our experiments.

The sensors communicate with the micro-controller via a proprietary 1-wire bus with power siphoned from the signal line-parasitic mode [18]. The sensor voltage and ground pins are joined to a common ground-plane. A unique 64-bit serial code identifies each sensor connected in an extended star network using twisted pair telegraph cables of a few meters in length. In Figure 3b, the sensors are shown with plastic rods attached that are used to strengthen the sensors for environmental sensing doors, although not fitted for calibration. The data signals from all sensors are multiplexed to a single wire along a cat5e Ethernet cable (5 m), while the other carries a common ground. Measurements were made simultaneously at 18-s intervals.

#### 2.3.3. Micro Controller

The Arduino–Yun combines a micro-controller environment and a version of Linux with a wireless stack. These and other components relevant to this experiment are shown in Figure 3c. Additionally, a prototype board is mounted on top that combines a real-time clock (RTC) for synchronization and a network connection (Ethernet) to multiplex the outputs of individual temperature sensors. The micro SD-card serves as the data backup which we access via the network. This experimental setup was fitted to a geometrical object and used for an eight month field trial remotely monitoring the object’s temperature [19].

## 3. Results

We investigated the accuracy, repeatability and nonlinearity (curvature) of the ds18b20 smart temperature device, an approach inspired by that of Meijer described in Ref. [5].

A representative temperature response for all sensors to the changing climate chamber is presented in Figure 4 showing the incremental temperature steps of 0.5 °C. As previously indicated in Section 2.3.1, we analyzed data recorded only within the plateau regions (5 °C intervals) from which we were able to derive the necessary offset between the sensor and reference temperatures for calibration. As mentioned before, there were approximately 20 min of stable measurement time for all reference temperatures except at 50 °C, which was slightly less.

Figure 5 shows a fit to a second-order polynomial curve representative] for the difference between sensor and climate chamber temperature measurements. It is clear that, even without further calibration, the sensor accuracy over the entire positive range is better than that stated in the ds18b20 data-sheet.

The variability of the 2nd order polynomial curve fits (Table 1) shows the mean and 3σ of the quadratic coefficients (Equation (Equation 7)),(7)$$f\left(T\right)=aT^{2}+bT+c,$$
where, *a*, *b*, and *c* represent the coefficients of the quadratic expression that best-fit the sensor data.

The mean temperature error curve and three times the standard deviation limits (3σ) for all sensors are shown in Figure 6. The curve fits are a minimum near 40 °C with a large variation in minima of ±6 °C. Note that the mean error curve is determined from the average value of all sensors at each calibration temperature and subsequently best-fitted to a second-order polynomial curve.

In Figure 7, we compare the mean offset error temperature for our dataset with those reported by the manufacturer with additional details described below, where the label indicates the appropriate sensors on the figure:Label **A** represents our results previously presented as the mean error curve in Figure 6.Label **B** results from the webpage ‘frequently asked questions for ds18b20 (FAQS: DS18B20)’ on the maxim integrated website, which shows the mean error for the ds18b20 sensor as a function of temperature and supply voltage [20]. There is a marginal difference in curvature with results and those of this paper showing a minima at the same reference temperature.Label **C**, for a ds1631 sensor [21], reported to be based on the same temperature device as the ds18b20 [22].Label **D** gives the result from the data sheet for the ds18b20 [16]. Observe that the minima are approximately 20 °C less than that of the other three, which presumably must be a difference in the foundry calibration temperature.

As a practical example of how the experimental method works (Section 2.2), we present the measurement and derived offsets for one of the sensors in Table 2. The offset temperature (T(offset)) represents the difference between reference and measured temperatures, from which the second order polynomial that best fits the offset values is calculated (Equation (Equation 7)). The remaining difference between the offset temperatures and the temperature calculated from the curve fit is T(res.).

The standard error on the mean of the measured sensor temperature is less than 0.02 °C with a 3σ variation of ±0.01 °C at reference temperatures all except for the highest. Note that the sensor mean standard error for the batch investigated is significantly smaller than the resolution increment (0.0625 °C at 12 bits).

The residual temperature errors after applying the calibration curves to our data are shown in Figure 8. This demonstrates the improvement in sensor accuracy after applying the curvature correction. The data points represent the mean values of all sensors, while the error bar shows the 3σ spread between sensors. The accuracy of all sensors is less than ±0.15 °C.

## 4. Discussion

What is meant by a precise temperature measurement will in part depend on the application, but typically an accuracy of order ±0.1 °C is reasonable for smart sensors. As demonstrated in this paper, and those referenced in this article, the temperature behavior of bi-polar transistors is very predictable and stable over time, albeit it is not exactly linear. We have shown that a calibration post-packaging such as curvature correction can significantly improve the accuracy of a smart sensor beyond that stated in the datasheet (Section 2.1).

From our results in Figure 5 and Figure 6 and as suggested by the manufacturer [23], a single point calibration is not sufficient if one wishes to exploit the potential of higher accuracy from a smart sensor with a predictable temperature response. Multiple calibration temperatures may be necessary, but this could result in further manufacturing cost. For example, post-packaging calibrations are required to adequately compensate circuits trimmed during the foundry stage that are then subjected to induced stress during high temperature packaging. A thermal stress may result in additional nonlinear temperature errors [24], which, for low-cost smart sensor calibration post-packaging, is time-consuming and adds additional costs [7]. This is the principal reason why only one reference temperature is used to calibrate often at the wafer stage before dicing into individual devices, which causes a stress-induced change in the resistant due to stress.

Our approach throughout this study is to investigate three aspects of the ds18b20 sensor’s temperature performance: (1) accuracy of our results with respect to the manufacturer’s specification, and comparison with other research data; (2) repeatability of measurements at a fixed and stable reference temperature over a short time period, and (3) the extent to which a nonlinear curvature calibration post-packaging can improve the temperature accuracy.

### 4.1. Accuracy

The temperature accuracy of a low-cost smart sensor, such as the one studied in this paper (ds18b20), will depend on the temperature range. This is aptly demonstrated both by the results in this paper and those from the data-sheet of the supplier [16]. Implicitly, we assume that a sensor that does not have on-chip compensation circuitry and was calibrated at a single foundry temperature, which results in a nonlinear curvature offset as shown in Figure 1b. The reported accuracy is ±0.5 °C within the temperature range investigated in this article (−20 °C to 50 °C), rising to ±2 °C for the full sensor range (−55 °C to 125 °C).

Our results (Figure 6) lay within the manufacturer’s specifications even though the exact curves are not totally consistent with those of the data-sheet. The temperature value of the curve minima for our results is nearly twice that shown in the datasheet, which implies a foundry calibration temperature of approximately half that of our results. Additional reported data for the same sensor also supports our results contradicting those from the datasheet [20]. Even the manufacturer’s specification curves for 3σ contradict its own limits at the highest values of 85 °C given that the projected value on the graph reaches only 70 °C. Additionally, the 3σ curves of the calibration offsets (Figure 7) show a narrowing (‘bottle-neck’) near the curve minimum, as expected from theory (Figure 1b). The bottle-neck occurs from 30 °C to 50 °C and again is not consistent with the manufacturer’s data sheet; however, our results do seem more consistent with the overall premise that the minima occurs at the temperature used in the foundry for trimming.

A comparison of the sensor accuracy of previous work shows an accuracy of between 0.1 °C to a few degrees for the full temperature range of the sensor [12]. Similar plastic-packaged devices can achieve an inaccuracy of ±0.15 °C (3σ) over a reduced range (−20 °C to 105 °C), while it is reported that a low-stress ceramic package can achieve inaccuracies of ±0.1 °C (3σ) after only one-point calibration [8,25]. An alternative method used by Yousefzadeh [26] achieved an accuracy comparable to conventional measurements using electrical trimming of the packaging at room temperature.

An additional consideration is how the sensor may respond to dynamic changes in the local temperature. The heat transfer across the sensor packaging (casing) will determine how quickly the energy reaches the sensor. For example, the ds18b20 is commonly available either in a molded plastic case or encapsulated in a metal tube. Ultimately, these devices are limited to how fast they can readout their data out along a network, which is dependent on the temperature conversion time and the number of sensors in the network (traffic). From 12-bit to 9-bit temperature data, the conversion time drops from 750 ms to 93.75 ms. Hence, one must consider both the dynamics of the signal together with the resolution limitations of the device.

### 4.2. Repeatability

The sensor’s ability to output the same temperature value while measuring a fixed climate chamber over a period of time is defined as its repeatability. This assumes that the variability of the climate chamber is significantly smaller than that of the packaged sensor. Importantly, the values we obtain are significantly lower than the level of ±0.02 °C.

Our results demonstrate that temperature measurements of the climate chamber are repeatable within a standard error on the mean of approximately three times less than that of the sensor at its most sensitive setting of 12 bits, i.e., a standard of error on the mean of approximately 0.02 °C for a sensor resolution of 0.06 °C. Consequently, we can be certain that our measurements closely predict the sample mean. It is important to note that these results show little significant variation with reference temperature. The only exception is at 50 °C, where we did not allow sufficient time for the chamber to fully stabilize.

As touched on before, the impact of packaging is an important consideration since plastic has a significantly lower thermal conductivity (≈0.23 W/(m · K)) compared to aluminum at 100 times larger. Consequently, any small chamber fluctuation due to changes in convection or radiation will be damped. Note that the long-term stability of the silicon temperature sensor is usually sufficient to keep its specification during its lifetime [25]. The measurements are not only very repeatable, but the spread between sensors is very small (less than a standard error on the mean of 0.01 °C).

### 4.3. Calibration

It is logical that the one foundry reference temperature used for trimming should be in the middle of your expected measurement range, implying approximately 40 °C for the ds18b20 as shown in our results, and not half that as suggested by the datasheet. In comparison, Bakker [25] considers an electronic approach to compensate for curvature. Their aim was to produce a precision sensor requiring only a single point calibration reference. Our results for the temperature offset show a very predictable quadratic variation; hence, calibration for nonlinear effects using multiple reference temperatures was a sensible option. In practice, the number of required reference temperatures can be reduced significantly, especially if the exact temperature of calibration during fabrication can be inferred. We showed a three-fold improvement in sensor accuracy across the full range of temperatures investigated after calibration at multiple references (−20 °C to 55 °C).

In particular, we assume a parabolic shaped curve with a minima at the temperature calibrated at fabrication; however, it is not strictly necessary to calibrate for many points used in this paper since any additional one to that of the fabrication temperature will fix a correction curve improving the sensor accuracy. Interestingly, this additional calibration does not necessarily need a climate controlled chamber since one could use a characteristic temperature transition such as the melting point of water.

It is likely that the plastic cased sensors suffer more mechanical stress than ceramic ones, where the stress is unlikely to be uniform across the whole batch of sensors [7]. Individual calibration will be necessary across the likely sensing range. As a consequence, if a temperature accuracy is required that is significantly higher than that quoted in the manufacturer’s specification, an external calibration using a number of temperatures is essential.

### 4.4. Limitations

The low number of sensors studied makes it difficult to fully appreciate the extent of the sample and batch variability—in particular, since our findings are fully consistent with those of the manufacturer. However, in terms of individual sensor calibration, this is not of concern. The climate chamber has an accuracy similar to the level we wish to achieve for the sensors, whereas a reference temperature should necessarily be of greater accuracy. Additionally, it is not understood whether the sensor operated in a parasitic power mode would respond as one with external power. Note that the use of parasitic power is not recommended for temperatures above 100 °C since the sensor may not be able to sustain communications due to higher leakage currents—consequently, the power version is necessary.

## 5. Conclusions

We have presented a calibration technique for smart temperature sensors based on curvature correction at multiple reference temperatures, which is shown to improve the accuracy of a ds18b20 device by nearly three-fold over a significant range of temperatures (−20 °C to 50 °C). The variation in difference between sensor and reference temperatures is well matched to a second order polynomial curve where each sensor exhibits slight differences in curvature. Our experiment demonstrates that the repeatability of temperature, with respect to the references used, is limited by the resolution of the analogue-to-digital converter and not fluctuations in the device. The residual sensor accuracy after curvature correction was reduced to approximately ±0.15 °C from that reported of ±0.5 °C across the entire range of measured reference temperatures (−20 °C to 50 °C). Finally, we observe that significantly less additional calibration points would be necessary to characterize the second order correction curvature, in principal only one extra, but this would require knowledge of the foundry calibration temperature and its offset.

## Figures and Tables

**Figure 1 sensors-18-04087-f001:**
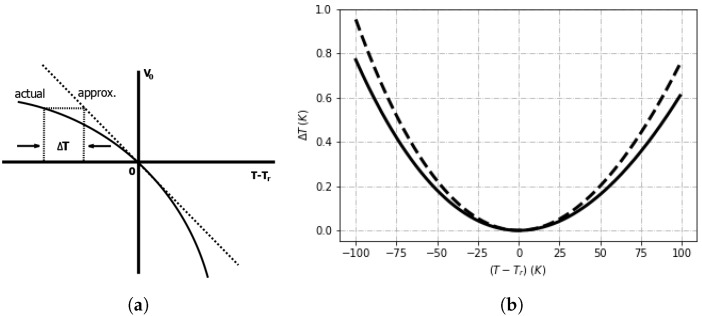
The nonlinear temperature error due to the curvature ($c(T-T_{r})$) of the $V_{BE}$: (**a**) T represents the deviation of the measured temperature from its linear approximation at a reference temperature $\left(T_{r}\right)$; (**b**) the curvature term is expressed as a temperature, and plotted for η=3.0 (solid line) and η=3.5 (dashed line).

**Figure 2 sensors-18-04087-f002:**
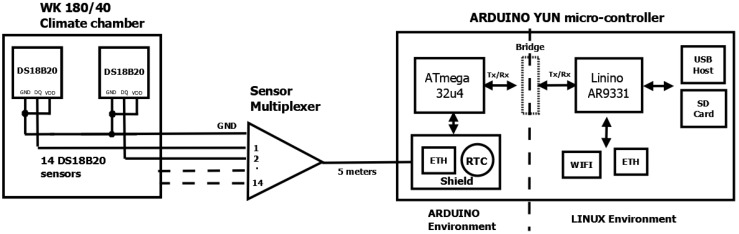
A schematic of the experimental arrangement, highlighting the key components: ds18b20 sensors within a climate chamber, sensor multiplexing and Arduino Yun micro controller.

**Figure 3 sensors-18-04087-f003:**
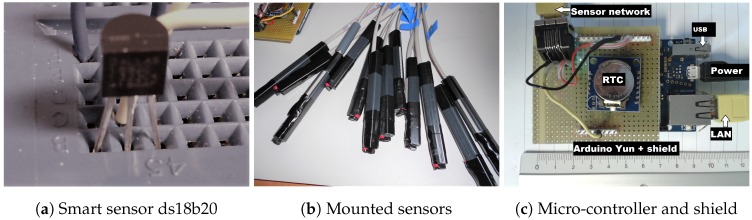
Images of the key experimental components: (**a**) ds18b20 sensor, (**b**) the sensors connected to network cables and mounted to plastic rods and (**c**) an Arduino–Yun micro controller with prototype board.

**Figure 4 sensors-18-04087-f004:**
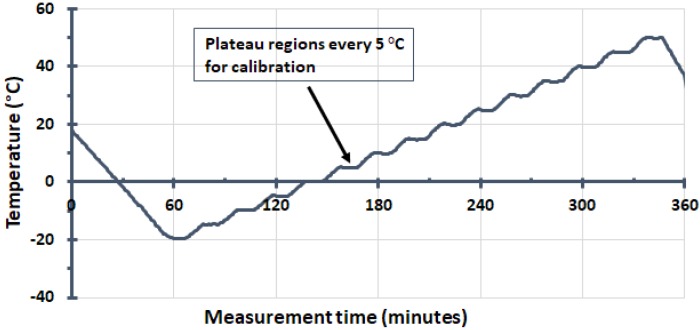
An example of the temperature profile of the climate chamber, measured by one of the sensors, used to calibrate the ds18b20 sensors. This plot is quite typical for all of the sensors investigated, with measurements recorded every 18 s. The duration of each 5 ${}^{{}^{\circ}}$C horizontal plateau region is approximately 20 min.

**Figure 5 sensors-18-04087-f005:**
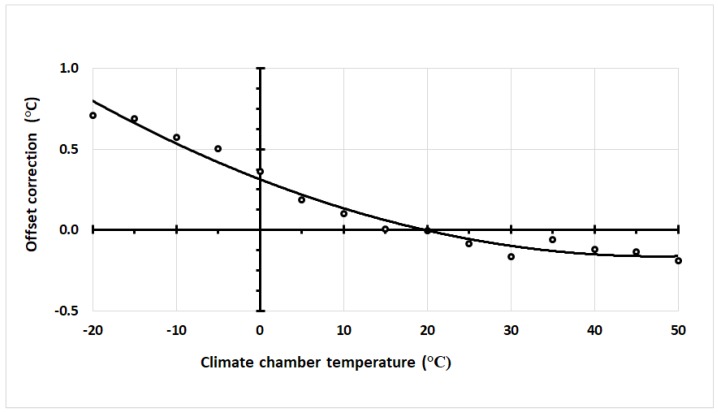
An example of the temperature offset necessary to correct the measured ds18b20 sensors for higher-order effects (o), compared to the best-fitting quadratic equation (solid line) to this data.

**Figure 6 sensors-18-04087-f006:**
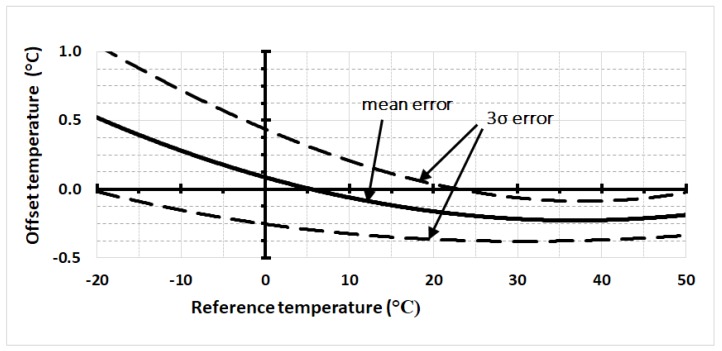
The calibration offsets for the batch of fourteen sensors is shown, where the solid line represents the mean offset error for the batch. The dashed lines are the 3σ confidence limits.

**Figure 7 sensors-18-04087-f007:**
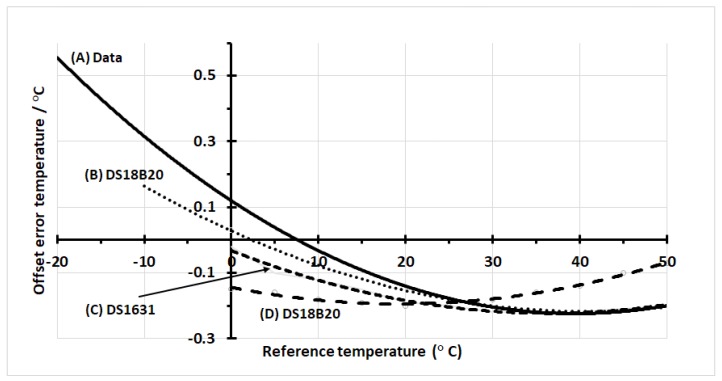
We show a comparison of the mean sensor offset errors with the reference temperature for our data (**A**) and from relevant manufacturer’s results (**B**–**D**). Plots A, B and D are from a ds18b20 sensor, while C is from a ds1631 sensor, which is an equivalent temperature sensor to that of a ds18b20.

**Figure 8 sensors-18-04087-f008:**
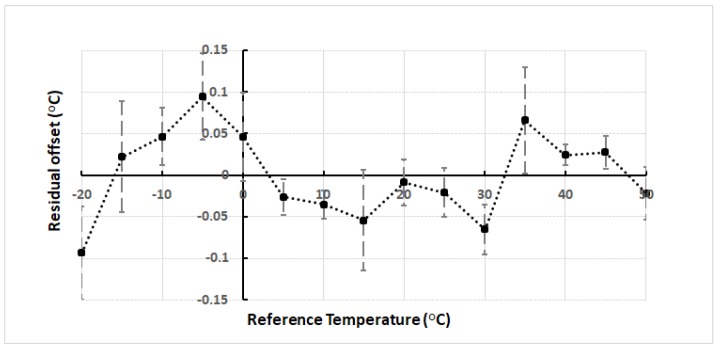
The residual error after application of the offset corrections to the measured temperature data is shown.

**Table 1 sensors-18-04087-t001:** Coefficients for the best-fitting second-order polynomial to all our sensor data. Both the mean value for the sensors and 3σ values are shown.

Coefficients	a	b	c
**Mean curve**	$2.2\times 10^{-4}$	$-1.74\times 10^{-2}$	0.12
3σ	$\pm 7.5\times 10^{-5}$	$\pm 9.8\times 10^{-3}$	±0.34

**Table 2 sensors-18-04087-t002:** An example of the method working for one of the sensors, showing the reference temperature (T(ref.)) and the measured data (T(data)) in degrees Celsius. The difference between the previous two represents the temperature offset (T(offset)=T(ref.)−T(data)). The final row T(res.)) shows that, after fitting the T(offset) values to a 2nd order polynomial, there is still a small residual offset.

**T (ref.)**	−20	−15	−10	−5	0	5	10	15	20	25	30	35	40	45	50
**T (data)**	−19.45	−14.44	−9.54	−4.60	0.25	5.08	9.99	14.90	19.90	24.84	29.75	34.88	39.83	44.85	49.82
**T (offset)**	−0.55	−0.56	−0.46	−0.40	−0.25	−0.08	0.01	0.10	0.10	0.16	0.25	0.12	0.17	0.15	0.18
**T (res.)**	0.10	−0.04	−0.06	−0.11	−0.06	0.02	0.03	0.05	0.00	0.00	0.05	−0.11	−0.08	−0.11	−0.08

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
