# Peer review of "Enhanced Accuracy of CMOS Smart Temperature Sensors by Nonlinear Curvature Correction"

_sensors, 2018, doi:10.3390/s18124087_

Reviewer 1 Report

Your paper entitled “Enhanced accuracy of CMOS smart temperature sensors by non-linear curvature correction” is very interesting.  The industrial and medical communities can benefit from CMOS smart temperature sensors with much improved accuracy.  Your unique approach to improve the accuracy of a low-cost digital smart temperature sensor (ds18b20) by correcting for the non-linear response using post-foundry calibration at multiple temperature references in a climatically controlled environment represents an advanced technique.

My comments/recommendations follows:

- Recommend that η (a temperature exponent in the analytical expression for saturation current) be defined prior to presenting Figure 1.  Please place Figure 1 after Equations 4-6.

- Recommend that all the Figures be referenced (referred to) before they appear in the document.

- What is the duration of your 5°C plateaus in Figure 4?

- Recommend that your accuracy of +0.15°C, which is mentioned in the Conclusion, also be mentioned in the Abstract.  It is not clear whether your accuracy of +0.15°C represents the full sensor range (-55°C to 125°C).

- Recommend that your accuracy of +0.15°C be compared to the reported accuracy of +0.5°C in the Conclusions and Abstract.

- In the "Accuracy" Section the following statement is given “The reported accuracy is +0.5°C within the temperature range investigated in this article.” Recommend that the temperature range investigated be restated here.

I made a few suggestions to improve sentence structure (See the attached PDF document – Comments / Recommendations are in the yellow sticky notes

Author Response

Firstly, the authors wish to thank the reviewer for his/her constructive and helpful comments concerning our paper.

Point 1: Recommend that η be defined prior to presenting fig 1. Please place fig 1 after equations 4-6

Response 1: Figure 1 placed after equation 4-6 between lines 107 and 108.

Point 2: Recommend that all figures be referenced (referred to) before they appear in the document

Response 2: Completed as suggested by the referee. Figures 2 and 3 are reference initially on page 3 and presented on pages 4 and 5. Figures 4, 5 and 6 are referenced on page 5 and presented on pages 7 and 8.

Point 3:  What is the duration of your 50C plateau in fig 4.

Response 3: The duration of the 50C plateau region is approximately 20 minutes, which now indicated on figure 4 (page 6) and within the text on pages 4 and 5.

Point 4:  Recommend that your accuracy of +/- 0.15 deg, which is mentioned in the conclusions also be mentioned in the abstract. It is not clear whether your accuracy of +/-0.15 deg represents the full sensor range (-55deg to 125deg)

Response 4: The measurement accuracy of +/-0.15 deg C is now mentioned in the abstract. We mention that this result is with respect to only a measured temperature range of between -22deg to +50deg.

Point 5:  Recommend that your accuracy of +/- 0.15 deg be compared to the reported accuracy of +/- 0.5 deg in the conclusions and abstract.

Response 5: The referee’s point is now reported in the conclusions as requested.

Point 6:  In the accuracy section the following statement is given,’ the reported accuracy is +/-0.5 deg within the temperature range investigated in this article,’ recommend that the temperature range investigated be restated here.

Response 6: As recommended, we have added the temperature range in the ‘accuracy’ section just after the reference to +/-0.5 deg.

Point 7:  A few suggestions to improve the sentence structure.

Response 7: The referee made a few suggestions to improve the sentence structure. We have included all these in the revised version of the text.

Thank you.

Reviewer 2 Report

This paper demonstrated a calibration method to improve the measurement accuracy of one kind of commercial thermal sensor. The method may be applied to improve the performance of other sensors, therefore it has merit for publication.

The authors presented photos of the actual devices. They should also present original measurement data, in particular the data for one or two practical cases to show how the method works.  

The authors should also discuss the applicable situations, e.g., does it work in dynamic cases where local temperatures change rapidly?

Author Response

Firstly, we wish to express our thanks for your constructive comments and suggestions to improve our paper.

Point 1: The author's present photos of the actual devices. They should also present original measurement data, in particular, the data for one or two practical cases to show how the method works.

Response 1: On your suggestion, we have added an additional table (table 2, page 8) describing the results for one of the sensors following the measurement method. Supporting text is given in the results section on page 6.

Point 2: The authors should also discuss the applicable situations e.g. does it work in dynamic cases where the local temperatures change rapidly?

Response 2: We included a paragraph in the discussion on the aspect of the sensors’ response to dynamic changes in the local temperature, which form part of an environmental experiment using these sensors in ref. 19. We mention important factors governing the dynamic response of the sensor (packaging, conversion time and network traffic).

Finally, we revised and improved the English in many of the sections in this paper, as per your suggestion.

Thank you

Reviewer 3 Report

The work presents series of experiments to improve the accuracy of a temperature sensor. They consist on calibrating the sensors and then employ non-linear correction terms to estimate the measured temperature.

There is an absolute lack of novelty in the approach. Every temperature sensor displays a certain degree of non-linearity in the transfer function. Adding higher-order polynomials to the approximated function surely improves the accuracy at the price of adding calibration points. Calibration is an expensive process that is energy and time consuming and requires the employment of a very precise thermal chamber.

The paper explains that this methodology is useful for low-cost smart temperature sensors that only have been calibrated at one point. However, calibrating again the sensors increases their overall cost significantly and it is very probable that the improvement in accuracy can be surpassed investing in better sensors in the first place.

English can be improved throughout the whole document.

Taking into account these considerations, I cannot recommend this paper for publication.

Author Response

Firstly, we wish to thank the reviewer for his/her honest assessment of our paper.

Point 1: There is an absolute lack of novelty in the approach. Every temperature sensor displays a certain degree of non-linearity in the transfer function. Adding higher-order polynomials to the approximated function surely improves the accuracy at the price of adding calibration points. Calibration is an expensive process that is energy and time consuming and requires the employment of a very precise thermal chamber.

Response 1:  The reviewer is perfectly correct to mention that additional calibration is an expensive process at the foundry. Consequently, we have added further text to reflect this important point in the introduction and discussion. However, we also make it clear that additional calibration is foreseen by the user and not the manufacturer, and may not necessarily require a precise thermal chamber, such as the one used in this case. Note, we investigate only second-order effects as shown in the theory in the materials and methods section.  

Point 2: The paper explains that this methodology is useful for low-cost smart sensors that only have been calibrated at one point. However, calibrating again the sensors increases their overall cost significantly and it is very probable that the improvement in accuracy can be surpassed by investing in better sensors in the first place.

Response 2: The reviewer is correct to point out our seemingly inconsistent approach of performing additional calibration to low-cost sensors compared to purchasing more accurate sensors in the first place. The key point here is that the sensors investigated (ds18b20) are very common in research and academic projects, and are often selected based on a compromise between accuracy and the additional benefits e.g. networkable, parasitically powered... The material section contains text describing these sensors and figure 2 shows a typical network set up. We wish only to show that under these conditions the same sensors could benefit significantly from additional calibration.

Point 3: English can be improved throughout the whole document.

Response 3: As suggested, we have revised the English in significant throughout the paper.

Thank you
